



# Regionalization of hydrological model parameters using gradient boosting machine

Zhihong Song[1,2], Jun Xia[1,2,3], Gangsheng Wang[1,2], Dunxian She[1,2], Chen Hu[1,2], Si Hong[1,2]

[1]State Key Laboratory of Water Resources and Hydropower Engineering Science, Wuhan University,
5    Wuhan, 430072, China
[2]Hubei Key Laboratory of Water System Science for Sponge City Construction, Wuhan University,
Wuhan, 430072, China
[3]Key Laboratory of Water Cycle and Related Land Surface Processes, Chinese Academy of Sciences,
Beijing, 10010, China

10    *Correspondence to*: Jun Xia (xiajun666@whu.edu.cn) and Gangsheng Wang (wanggs@whu.edu.cn)





**Abstract.** Regionalization of hydrological model parameters is key to hydrological predictions in ungauged basins. The commonly used multiple linear regression (MLR) method may not be applicable in complex and nonlinear relationships between model parameters and watershed properties. Moreover, most regionalization methods assume lumped parameters for each catchment without considering within-catchment heterogeneity. Here we incorporated the Penman-Monteith-Leuning (PML) equation into the Distributed Time-Variant Gain Model (DTVGM) to improve the mechanistic representation of the evapotranspiration process. We calibrated six key model parameters grid-by-grid across China using a multivariable calibration strategy, which incorporates spatiotemporal runoff and evapotranspiration (ET) datasets (0.25°, monthly) as reference. In addition, we used the gradient boosting machine (GBM), a machine learning technique, to portray the dependence of model parameters on soil and terrain attributes in four distinct climatic zones across China. We show that the modified DTVGM could reasonably estimate the runoff and ET over China using the calibrated parameters, but performed better in humid than arid regions for the validation period. The regionalized parameters by the GBM method exhibited better spatial coherence relative to the calibrated grid-by-grid parameters. In addition, GBM outperformed the stepwise MLR method in both parameter regionalization and gridded runoff simulations at national scale, though the improvement is not significant pertaining to watershed streamflow validation due to most of the watersheds being located in humid regions. We also revealed that the slope, saturated soil moisture content, and elevation are the most important explanatory variables to inform model parameters based on the GBM approach. The machine-learning-based regionalization approach provides an effective alternative to deriving hydrological model parameters by using watershed properties in ungauged regions.

**Keywords**: Regionalization; Gradient Boosting Machine; Distributed hydrological model; Soil; Terrain;



## 1 Introduction

Hydrological modeling can provide quantitative extrapolation or prediction of runoff and water balance (Beven, 2011; He et al., 2011), which serves as the basis for water management for human livelihood, agriculture, industry, and environment (Hobeichi et al., 2018; Montanari et al., 2013; Parajka et al., 2013; Zhang et al., 2020) (Bao et al., 2012; Guo et al., 2021; Mizukami et al., 2017; Pagliero et al., 2019; Randrianasolo et al., 2011; Yang et al., 2020). Hydrological models often require streamflow and/or other

observations to calibrate parameters (Beck et al., 2020). However, it is difficult to parameterize a hydrological model at large scales (e.g., from national to global) or remote regions due to the sparse or the lack of observing stations. Under such circumstances, attempts have been made to use reanalysis datasets, not the observations, for model calibration and validation (Bai et al., 2018; Dembélé et al., 2020; Huang et al., 2020; Immerzeel and Droogers, 2008; Zhang et al., 2020). For example, Dembélé et al.,

(2020) developed a novel multivariate calibration framework on spatial patterns by combining streamflow with satellite datasets, including evapotranspiration (ET), soil moisture, and terrestrial water storage to parameterize a distributed hydrological model. Zhang et al., (2020) and Huang et al. (2020) demonstrated encouraging potential in the calibration of hydrological models solely against remote sensed ET data (or bias-corrected remote sensed data) without the need for observed streamflow data (i.e., the runoff-free

calibration approach) to predict runoff in ungauged basins. The current study attempted to use spatiotemporal ET and runoff data for grid-by-grid calibration of a distributed hydrological model.
Hydrological models generally rely on regionalization methods to tackle the Predictions in Ungauged Basins (PUBs) by transferring information from gauged to ungauged catchments (He et al., 2011; Parajka et al., 2013; Razavi and Coulibaly, 2013). Regionalization of hydrological parameters generally includes

three categories, similarity-based, regression-based, and hydrological signatures-based (Guo et al., 2021). The similarity-based regionalization presumes that the catchments with similar characteristics have the same hydrological response, such as the spatial proximity (Oudin et al., 2008; Parajka et al., 2005; Samuel et al., 2011; Vandewiele and Elias, 1995), and the physical similarity (Beck et al., 2016; Oudin et al., 2010; Yang et al., 2018; Zhang and Chiew, 2009). The regression-based method aims to establish the

regression relationship between hydrological parameters and catchment characteristics (e.g., soil, topography, and climate variables), which helps to estimate model parameters in ungauged regions



(Hundecha and Bárdossy, 2004; Livneh and Lettenmaier, 2013; Xu, 1999; Young, 2006). In addition, some researchers have managed to transplant hydrological signatures (runoff depth, runoff ratio, flow percentile, flood frequency, baseflow index, flow change rate, etc.) from gauged catchments to ungauged basins (Castiglioni et al., 2010; Oubeidillah et al., 2014; Yang et al., 2019).

Among the most utilized regionalization techniques are probably regressions between the model parameters and physiographic catchment attributes as they are simple, fast, and intuitive (Bao et al., 2012; Heuvelmans et al., 2006; Oudin et al., 2008; Pagliero et al., 2019; Parajka et al., 2005; Razavi and Coulibaly, 2013; Young, 2006). Typically, the multiple linear regression (MLR) is widely used to estimate model parameters (Pagliero et al., 2019; Parajka et al., 2005; Sefton and Howarth, 1998). However, there are several limitations for regression approach, including less representative results of linear regression due to multicollinearity in catchment attributes, a high correlation between explanatory variables, complex and nonlinear relationships with high nonstationarity between physical catchment descriptors and model parameters (Blöschl, 2005; Guo et al., 2021; Kuczera and Mroczkowski, 1998; Pagliero et al., 2019; Yang et al., 2020; Zhang et al., 2018).

Machine learning techniques provide an alternative to overcome these issues for the linear regression approaches. For example, Sun et al., (2014) found that the Gaussian process regression is superior to traditional linear regression and artificial neural network models in most cases for probabilistic streamflow forecasting in 438 catchments across the United States. Zhang et al., (2018) assessed the regression tree ensemble approach compared with MLR, log-transformed MLR, and hydrological modeling in 605 catchments across Australia, which outperforms the two linear regressions in predicting signatures of flow dynamics. Prieto et al., (2019) implemented a machine learning technique, i.e., random forests, combined with a Bayesian inference formulated for regionalized principal components of a set of flow indices in 92 catchments in northern Spain. Heuvelmans et al., (2006) demonstrated that artificial neural networks could provide a useful alternative in some cases compared with linear regression, especially when we could physically explain the nonlinear relationship between parameters and catchment descriptors.

Here we attempted to estimate parameters as a function of watershed features in ungauged areas using a machine learning method, i.e., the gradient boosting machine (GBM) (Friedman, 2001). GBMs are a



family of powerful machine-learning techniques that have achieved considerable success across many domains, such as image classification (Lawrence, 2004), text classification (Natekin and Knoll, 2013), pattern recognition (Schütz et al., 2019), motion detection (Bouwman et al., 2020), and ecological and environmental issues (Fan et al., 2021; Liao et al., 2020; Wei et al., 2019; Xia et al., 2020). The principal idea behind GBM is to consecutively construct the new base-learners, which optimally improves

prediction in combination with the already existing ensemble leading to a more accurate estimate of the response variable (Natekin and Knoll, 2013; Schütz et al., 2019). We used the MLR to evaluate and compare the effectiveness of GBM for parameter regionalization.

We incorporated the Penman-Monteith-Leuning (PML) equation (Leuning et al., 2008) into the Distributed Time-Variant Gain Model (DTVGM) (Wang et al., 2009; Xia et al., 2005) forced with state-

of-the-art meteorological data to predict runoff and ET over China. Our specific objectives were to (i) explore the capability of GBM for parameter regionalization relative to the traditional MLR method; (ii) develop China-wide hydrological parameters that are linked to soil and terrain properties; and (iii) identify the critical factors for these hydrological parameters in distinct climatic zones.

## 2 Materials and Methods

We ran the China-wide hydrological model in a spatially distributed fashion to account for within-catchment heterogeneity, the discrepancy in scale, and thus rainfall-runoff behavior between catchments and grid cells (Beck et al., 2020). Figure 1 presents an overview of this study. We first conducted a multivariable calibration to derive an ensemble of grid-by-grid parameter maps for regionalization. The multivariable datasets for model calibration and validation included a runoff product (Zhang et al., 2014)

and an evapotranspiration (ET) dataset (Martens et al., 2017). The runoff product (0.25°×0.25°) was derived from the Variable Infiltration Capacity (VIC) model where the simulated monthly streamflow matches well with the measurements at the major river basins in China (Zhang et al., 2014). We obtained the total runoff depths from the dataset for model calibration and validation. The ET product used in this study was the 0.25°-resolution Global Land Evaporation Amsterdam Model (GLEAM) V3.3a dataset

(Martens et al., 2017), which has been demonstrated to estimate actual ET with reasonable accuracy in China (Yang et al., 2017).



We regionalized model parameters using the MLR and GBM methods in terms of the explanatory variables, including topographical and soil characteristics. Finally, we compared the model performance with parameters from grid-by-grid calibration and regionalization.

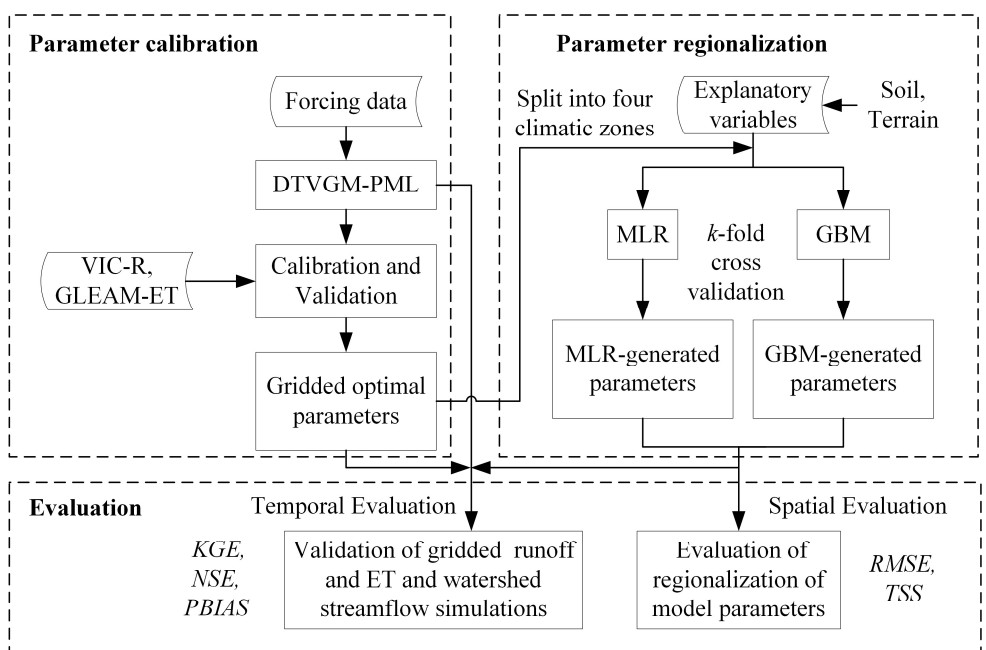

**Figure 1. Flowchart of parameter calibration, validation, and regionalization. DTVGM-PML: the distributed time-variant gain model with the penman-monteith-leuning equation; VIC-R: runoff simulated by the Variable Infiltration Capacity (VIC) model; GLEAM-ET: evapotranspiration dataset from the Global Land Evaporation Amsterdam Model (GLEAM) V3.3a; MLR: stepwise multiple linear regression; GBM: gradient boost machine; KGE: Kling-Gupta efficiency; NSE: Nash-Sutcliffe efficiency; PBIAS: percent bias; RMSE: root-mean-square error; TSS: Taylor skill score.**

## 2.1 China-wide hydrological model

We developed the DTVGM-PML model (Figure 2) in this study to implement China-wide hydrological modeling, which coupled the Penman-Monteith-Leuning (PML) equation and the distributed time-variant gain model (DTVGM). The DTVGM has been successfully applied to many river basins in China (Cai et al., 2014; Ning et al., 2016; Wang et al., 2009; Xia et al., 2005; Zeng et al., 2020; Zhan et al., 2013; Zou et al., 2017). We selected the model for its parsimonious model structure with limited free parameters due


to the lack of ground-based measurements (e.g., observed streamflow) for model calibration in many regions of China. We replaced the empirical evaporation model in the DTVGM with the PML equation
to improve the mechanistic representation of the ET process (Bai et al., 2018, 2020; Li et al., 2009; Zhang et al., 2008; Zhang et al., 2009; Zhou et al., 2013).We also coupled a snow routine from the HBV (Hydrologiska Byråns Vattenavdelning) model (Seibert and Vis, 2012) and the Gash rainfall interception model (van Dijk and Bruijnzeel, 2001) to improve relevant processes in DTVGM. The DTVGM-PML model ran on a grid-scale, with a spatial resolution of 0.25°×0.25°. The gridded runoff simulated by the
DTVGM-PML was routed by the Lohmann routing model for specific watersheds (Lohmann et al., 1996). In the PML equation, we estimated the ratio of soil evaporation to the equilibrium rate ($f$) using the relative soil water storage, $W/W_M$, simulated in the DTVGM. Another key parameter, the maximum stomatal conductance ($g_{sx}$), was assigned for each land cover type recommended by Zhang et al., (2017). Other insensitive parameters were held constant since ET simulations have no significant accuracy loss (Bai et
al., 2018; Leuning et al., 2008; Zhang et al., 2008).

In this study, we calibrated six parameters in the runoff generation process of the DTVGM-PML: two parameters ($g_1$, $g_2$) that control the nonlinear surface runoff generation, the subsurface runoff generation coefficient ($k_s$), the groundwater recharge coefficient ($k_r$) and recession coefficient ($k_g$), and the soil moisture storage capacity ($W_M$) (Figure 2). We provided detailed descriptions of the DTVGM-PML in
the supplementary materials.



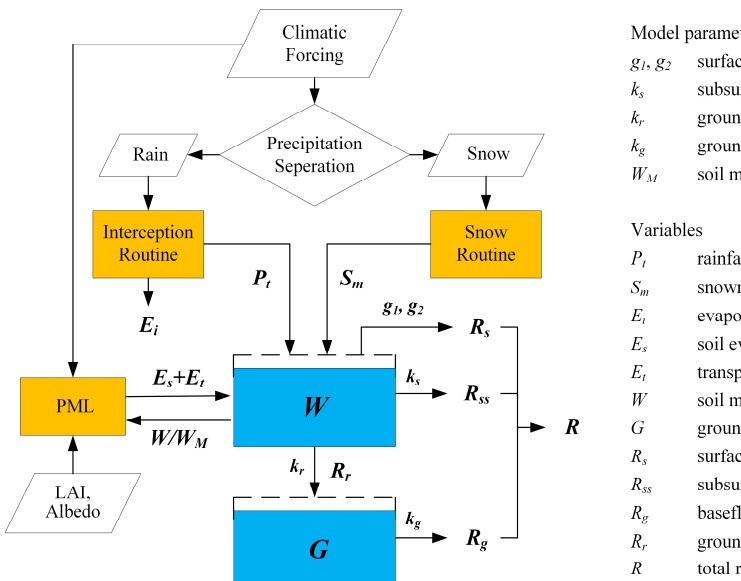

**Model parameters**

$g_1, g_2$    surface runoff generation coefficient (-)
$k_s$    subsurface runoff generation coefficient (-)
$k_r$    groundwater recharge coefficient (-)
$k_g$    groundwater runoff recession parameter (-)
$W_M$    soil moisture store capacity (mm)

**Variables**

$P_t$    rainfall passing through the canopy (mm)
$S_m$    snowmelt down to soil (mm)
$E_i$    evaporation of intercepted precipitation (mm)
$E_s$    soil evaporation (mm)
$E_t$    transpiration (mm)
$W$    soil moisture content (mm)
$G$    groundwater storage (mm)
$R_s$    surface runoff (mm)
$R_{ss}$    subsurface runoff (mm)
$R_g$    baseflow (mm)
$R_r$    groundwater recharge (mm)
$R$    total runoff (mm)

**Figure 2. The model structure of the DTVGM-PML.**

## 2.2 Parameter calibration strategy

In this study, we performed the 0.25° grid-by-grid calibration by fitting gridded monthly runoff and ET
data at a national scale (a total of 15640 grid-cells) owing to the limited long-term observed streamflow
data. The modeling period spanned from 1982–2012 and consisted of 15 years (1998–2012) of the
calibration period and 16 years (1982–1997) of the validation period. We used the Shuffled Complex
Evolution (SCE-UA) algorithm (Duan et al., 1992, 1994) for model calibration by minimizing a multi-
variable function (see Eq.(1)). The objective function is expressed as the Euclidean distance (denoted by
$F$) that combines the Kling-Gupta efficiency ($KGE$, see Eq. (2)) (Gupta et al., 2009; Kling et al., 2012) of
monthly runoff ($KGE_R$) and ET ($KGE_{ET}$) . The $KGE$ is a comprehensive criterion to measure the
agreement between observed and simulated values ranging from −∞ to 1, with an optimal value of 1.

$$F = \sqrt{w_1(1 - KGE_R)^2 + w_2\beta(1 - KGE_{ET})^2}, \tag{1}$$

$$KGE = 1 - \sqrt{(r - 1)^2 + (\beta - 1)^2 + (\gamma - 1)^2}, \tag{2}$$





where $w_1$, $w_2$ are the weights assigned to runoff and ET evaluation, respectively. In this study, $w_1$, $w_2$ were

both equal to 1. $r$ is the Pearson correlation coefficient, $\beta$ denotes the bias term (i.e., a ratio of means),

and $\gamma$ is the variability term (i.e., a ratio of coefficients of variation).

### 2.3 Parameter regionalization strategy

Regionalization techniques generally include two types, such as distance-based (spatial proximity,

physical similarity) and regression-based methods (He et al., 2011). In this study, we obtained the 0.25°

gridded parameters of the DTVGM-PML based on a multi-variable calibration, which were then divided

into four climatic zones over China (see Figure 3, such as humid, semi-humid, semi-arid, and arid region).

For each grid-cell, we estimated the relationship between the calibrated parameters (respond variable)

and physical properties (explanatory variables, e.g., topographical attributes and soil characteristics, see

Table 1) by a machine learning technique, the gradient boosting machine (GBM) (Friedman, 2001). For

comparison of the performance of GBM, we also examined the traditional regression method, the multiple

linear regression method (MLR) (Heuvelmans et al., 2006; Waseem et al., 2016), for parameter

regionalization as the benchmark.

The GBM is a powerful machine learning technique to train decision trees in a gradual, additive, and

sequential manner (Friedman et al., 2000; Friedman, 2001; Natekin and Knoll, 2013). The main idea of

the GBM is to add new models with respect to the error of the whole ensemble learned so far to the

ensemble sequentially to boosts its performance iteratively. The final GBM model is a stagewise additive

model of previous individual trees. The GBM has been proven successful across many domains, including

classification problems (Lawrence, 2004; Xia et al., 2020) and regression problems (Liao et al., 2020;

Xenochristou et al., 2020; Yan et al., 2019) which is the case for this study.

The MLR approach is a standard multiple linear regression to relate response variables to the explanatory

variables in a simple, fast and straightforward manner (Lima et al., 2015; Zhang et al., 2018). And the

stepwise selection of predictors is applied to minimize the possible errors resulting in the best performing

model and then identify the most influenced physiographical variable (Lima et al., 2015; Shu and Ouarda,

2012; Waseem et al., 2016). Unlike the GBM method, the MLR can explicitly quantify the relationship

between explanatory variables and response variables through a regression equation.





The GBM and MLR modeling were conducted using the 'gbm' and 'lmStepAIC' methods, respectively, with the *k*-fold cross validation in the R package 'caret' (Kuhn et al., 2020). The performing of the *k*-fold cross validation (*k*=10 in this study) can help to reduce the chances of overfitting, leading to less prediction variability and, therefore, improved accuracy (Natekin and Knoll, 2013). First, we considered two categories of representative explanatory variables for regression modeling in each grid-cell: (i) topographical variables, e.g., elevation (m), and slope (°); (ii) soil variables, e.g., sand content (g/kg), silt content (g/kg), clay content (g/kg), field capacity, wilting point, residual moisture content, saturated moisture content; saturated hydraulic conductivity (cm d$^{-1}$). Second, we eliminated the grid-cells with either $KGE_R$ or $KGE_{ET}$ less than zero, which perform poorly in simulating runoff or ET, and then split the remaining grid-cells into four subsets according to the climatic zones. Finally, we trained and evaluated the GBM and MLR modeling using the model parameters in each subset and the relevant explanatory variables.

### 2.4 Evaluation criteria

We used the Root-mean-square error (*RMSE*) to evaluate the performance of the parameter prediction based on the GBM and MLR modeling. A lower *RMSE* indicates better performance than a higher one. We also calculated the Taylor skill score (*TSS*) (Taylor, 2001) to express a synthetic measure of the prediction skill of the MLR and GBM modeling for model parameters. The *TSS* is a numerical summary of the Taylor diagram, varying from zero (least skillful) to one (most skillful). As defined in Eq.(3), the *TSS* increases monotonically with increasing correlation ($R \rightarrow R_0$) for any given variance, and increases as the modeled variance approaches the observed variance ($SDR \rightarrow 1$) for any given correlation. The Kling-Gupta efficiency (*KGE*) (Gupta et al., 2009; Kling et al., 2012), Percent Bias (*PBIAS*) (Gupta et al., 1999), and Nash-Sutcliffe efficiency (*NSE*) (Nash and Sutcliffe, 1970) were used for the evaluation of model simulations based on three parameter sets. The *PBIAS* varying from $-\infty$ to $+\infty$ measures the extent to which the simulated values are overestimated (a positive value) or underestimated (a negative value) relative to the observed values. The *NSE* is a widely used evaluation index to assess the predictive skill of hydrological models, which ranges from $-\infty$ to its perfect score, that is, 1. In addition to the KGE shown in Eq. (2), other evaluation criteria are formulated as follows:



$$RMSE = \sqrt{\frac{1}{n}\sum_1^n(X_{obs} - X_{sim})^2}, \tag{3}$$

$$TSS = \frac{4(1+R)^4}{\left(SDR \ \frac{1}{SDR}\right)^2 (1+R_0)^4}, \tag{4}$$

$$PBIAS = \frac{\sum_1^n(X_{sim}-X_{obs})}{\sum_1^n X_{obs}} \times 100\%, \tag{5}$$

$$NSE = 1 - \frac{\sum_1^n(X_{obs}-X_{sim})^2}{\sum_1^n(X_{obs}-\bar{X}_{obs})^2}, \tag{6}$$

where $X_{obs}$, $X_{sim}$ are the observed and simulated values, respectively. $\bar{X}_{obs}$ is the average observed value. $R$ represents the spatial correlation coefficient between the regionalized parameters and the calibrated

parameters, and $R_0$ is the maximum correlation attainable (0.999 in this study). $SDR$ is the ratio of the spatial standard deviation of the regionalized parameters against that of the calibrated parameters. $n$ is the total number of observations.

## 2.5 Data sources and processing

Data used in this study consisted of three categories, such as forcing data, attribute data, and evaluation

data (see Table 1).

(1) Model forcing data, including the climate forcing data and land surface data, were used for DTVGM-PML simulation. We used the China meteorological forcing dataset (CMFD) provided by National Tibetan Plateau Data Center (He et al., 2020; Yang et al., 2010; Yang and He, 2019), which contains seven daily variables (see Table 1) with a spatial resolution of 0.1°. The land surface variables used

here included leaf area index (LAI) and albedo obtained from the 8-day composite 0.05° × 0.05° GLASS product in National Earth System Science Data Center. The 8-day composite data were interpolated into the daily data using a piecewise cubic Hermite polynomial and then smoothed by the Savitzky-Golay filtering method (Fang et al., 2008; Li et al., 2009; Ruffin et al., 2008).

(2) Ten attribute variables were considered, including topographical attributes and soil characteristics.

The elevation and slope were derived from 90m digital elevation model. The soil data provided by





Liu et al., (2020) included eight variables summarized in Table 1 at multiple depths 0–5, 5–15, 15–30, 30–60, 60–100, and 100–200 cm. We transformed multiple-layer soil data into single-layer data using a weighted average method.

(3) In addition to the total runoff data from the VIC simulation for China and ET data from the GLEAM v3.3a product used for parameter calibration (section 2.2), the evaluation data also included observed daily streamflow from 31 representative watersheds (see Figure 3) for streamflow validation over China. Table S1 lists the basic information of the 31 hydrological stations.

Since the DTVGM-PML ran at a daily time step for 1982–2012 with a spatial resolution of 0.25°, all spatial data used in this study were resampled to 0.25° consistently.





**Table 1. Model Input for Model Simulation, Training, and Evaluation**

| Data type | Name | Unit | Sources |
|---|---|---|---|
| Forcing data | 2 m air temperature | K | http://data.tpdc.ac.cn |
| | Surface air pressure | Pa | Idem |
| | Specific humidity | kg kg$^{-1}$ | Idem |
| | 10 m wind speed | m s$^{-1}$ | Idem |
| | Downward shortwave radiation | W m$^{-2}$ | Idem |
| | Downward longwave radiation | W m$^{-2}$ | Idem |
| | Precipitation rate | mm hr$^{-1}$ | Idem |
| | LAI | m$^2$ m$^{-2}$ | http://www.geodata.cn |
| | Albedo | Unitless | Idem |
| Attribute data | Elevation (elev) | m | http://srtm.csi.cgiar.org/ |
| | Slope (slp) | ° | Estimated using elevation |
| | Sand content (snd) | g kg$^{-1}$ | (Liu et al., 2020) |
| | Silt content (slt) | g kg$^{-1}$ | Idem |
| | Clay content (cly) | g kg$^{-1}$ | Idem |
| | Field capacity (fc) | cm$^3$ cm$^{-3}$ | Idem |
| | Wilting point (pw) | cm$^3$ cm$^{-3}$ | Idem |
| | Residual moisture content (thr) | cm$^3$ cm$^{-3}$ | Idem |
| | Saturated moisture content (ths) | cm$^3$ cm$^{-3}$ | Idem |
| | Saturated hydraulic conductivity (ksat) | cm d$^{-1}$ | Idem |
| Evaluation data | Runoff | kg m$^{-2}$ | VIC simulation for China (Zhang et al., 2014) |
| | ET | mm month$^{-1}$ | GLEAM v3.3a product (https://www.gleam.eu/) |
| | Streamflow | m$^3$ s$^{-1}$ | China's Hydrological Year Book |

Note. The attribute variable abbreviation is shown in the parenthesis after the variable name.



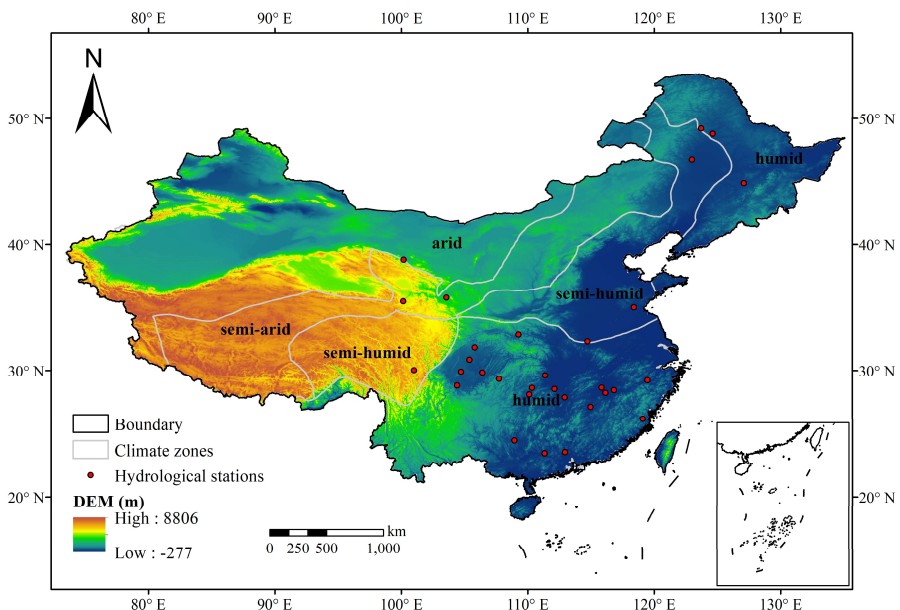

**Figure 3. Location of climatic zones (humid, semi-humid, semi-arid, and arid) and hydrological stations in China.**

## 3 Results

**3.1 Simulation of DTVGM-PML**

The model calibration set out using reference data (gridded monthly runoff and ET) from the former 16 years (1982–1997) or the latter 15 years (1998–2012) with the remaining period as the validation period. The model shows slightly better results when using data from the latter period than the first period for calibration (Figure S1), which was probably due to the better data quality in the latter period than in the

former period. Several previous studies have adopted this strategy, i.e., using data from the latter period for model calibration (Mizukami et al., 2017; Newman et al., 2017; Yang et al., 2018). Thus, the present study used the calibrated model parameters from the latter period for regionalization.

Figure 4 presents the spatial patterns of mean annual runoff and ET simulations derived from the DTVGM-PML during 1982–2012. Both the mean annual runoff and ET simulations show a decreasing





trend from southeast to northwest China, with the highest values in humid tropical and subtropical regions, intermediate values in temperate regions, and lowest values in cold and arid regions. Figure 5 shows the model performance of runoff and ET simulations in calibration and validation periods. The median *KGE* and *PBIAS* values for runoff simulation were 0.78 and 0.8%, respectively, in the calibration period, and 0.69 and −14.2%, respectively, in the validation period. The corresponding statistical values for ET

simulation were 0.70, −8.2% and 0.68, −13.1%, respectively. Overall, the DTVGM-PML could well simulate the monthly runoff and ET over China.

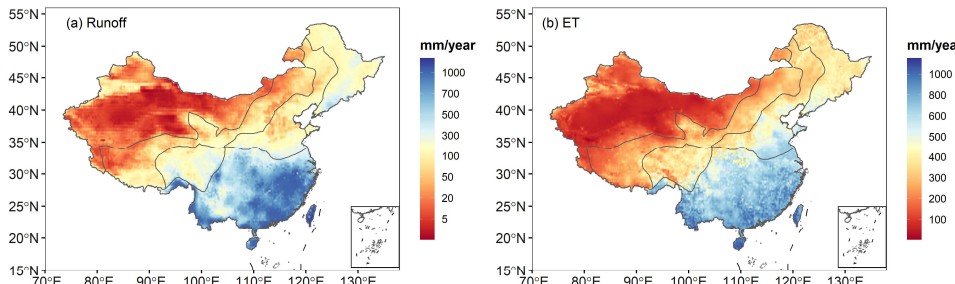

**Figure 4. Spatial patterns of mean annual runoff (a) and ET (b) simulations by the DTVGM-PML during 1982–2012.**




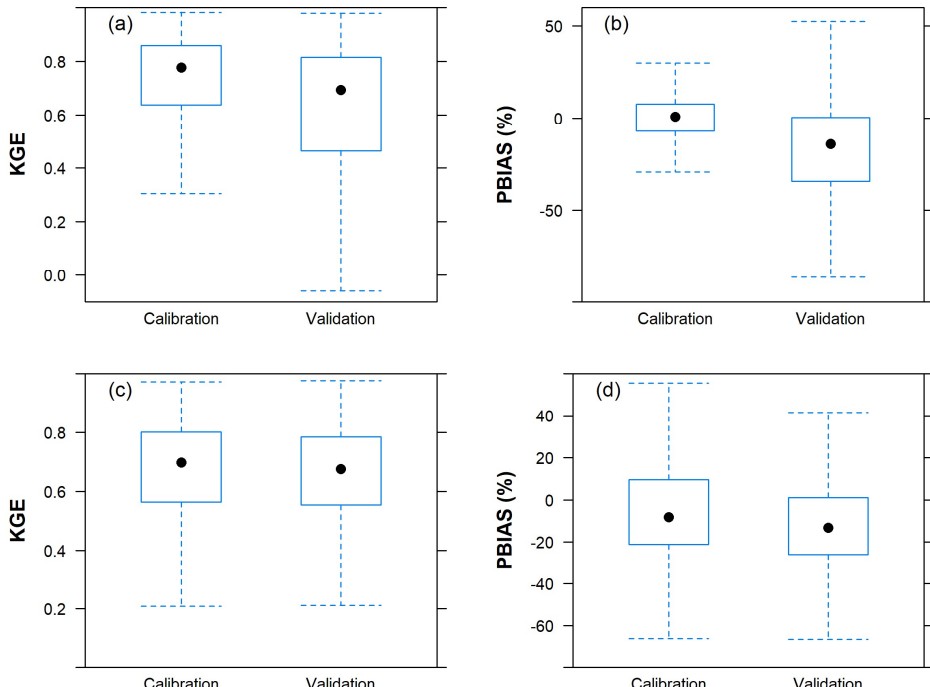

**Figure 5. Model performance of runoff (*KGE*: a, and *PBIAS*: b) and ET (*KGE*: c, and *PBIAS*: d) simulations in calibration and validation periods. The boxplot was generated using data from a total of 15640 grid cells over China.**

### 3.2 Regionalization of model parameters


We evaluated the regionalization model performance in terms of RMSE for six parameters in four climatic zones. As shown in Figure 6, GBM appears better in predicting model parameters than MLR because of the lower RMSE for all parameters in humid regions. We found consistently better accuracy of GBM in semi-humid, semi-arid, and arid areas (Figure S2 to Figure S4). Additionally, the difference in model


performance between GBM and MLR was significant (p-value < 0.05) as per the Kruskal-Wallis test (Hollander et al., 2013). Overall, these results suggest that the performance of GBM is significantly better than that of MLR for six parameters in four climatic zones. We also calculated the Taylor skill scores



(*TSS*) of the two regionalization models in predicting model parameters across China (Figure S5). There was a significant improvement of the GBM with higher *TSS* values than the MLR in prediction for model

parameters.

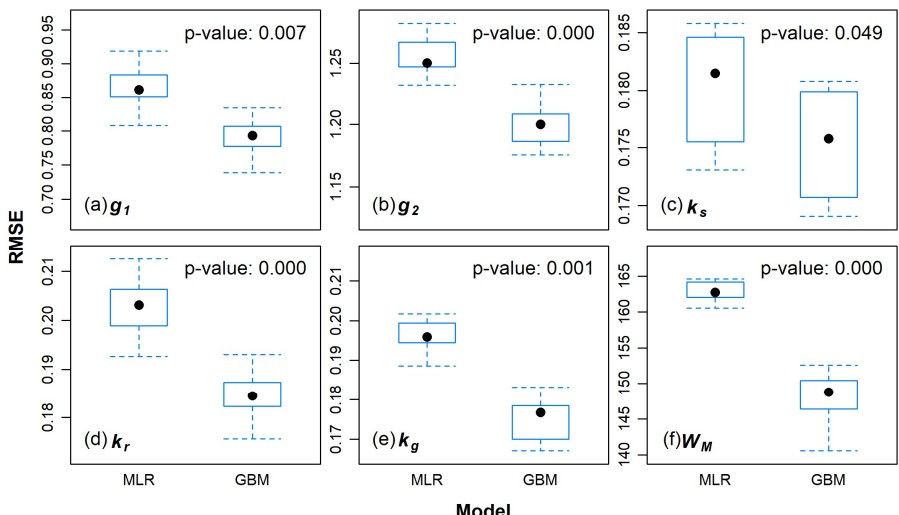

**Figure 6. Performance evaluation of MLR and GBM for six parameters, (a) $g_1$, (b) $g_2$, (c) $k_s$, (d) $k_r$, (e) $k_g$, (f) $W_M$, in humid region. MLR and GBM denote the multiple linear regression with stepwise selection and the gradient boosting machine model. The boxplot is generated from the 10 samples in *k*-fold cross validation. We use the non-**
**parametric Kruskal-Wallis (KW) test to determine the significance of difference in the performance between MLR and GBM at a significance level of 0.05.**

Figure 7 shows the spatial patterns of the three parameter sets derived from calibration and regionalization (MLR and GBM). Generally, both MLR and GBM derived parameters exhibited good agreement spatially

with the calibrated parameters. As the model parameters were related to topography and soil properties, the parameters generated by MLR and GBM show exquisite spatial patterns and a much better spatial coherence than the calibrated parameters. Compared with the MLR generated parameters, the GBM generated parameters presented more consistency with the calibrated parameters in space. For example (Figure 7 e1, e2, and e3), the MLR underestimated the parameter $k_g$ in part of Western China (nearly

0.25–0.5) relative to the calibrated parameters (about 0.5–0.8), while the GBM-derived parameters (0.5–





0.75) were more consistent with calibrated values. In summary, the regionalized parameters generated by the regionalization methods (MLR and GBM) exhibited better spatial coherence relative to the calibrated parameters with spatial discontinuities. The GBM derived more accordant parameters with the calibration than the MLR.



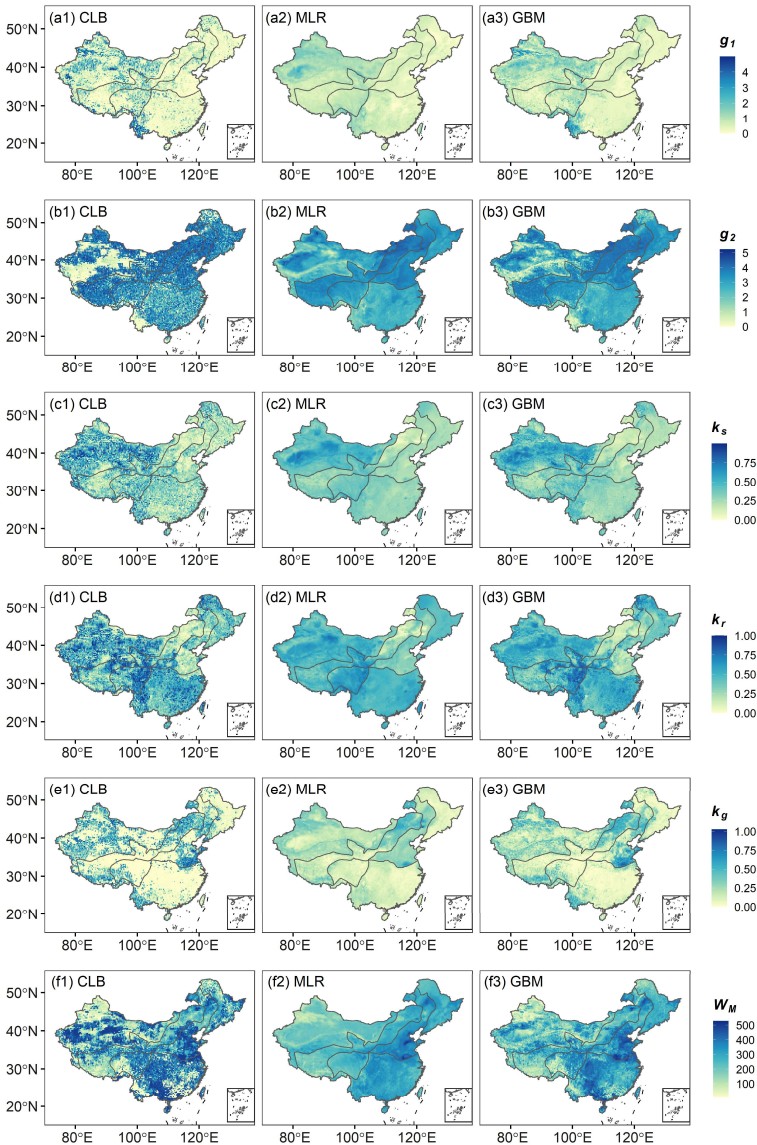


**Figure 7. Spatial patterns of model parameters: (a) $g_1$, (b) $g_2$, (c) $k_s$, (d) $k_r$, (e) $k_g$, (f) $W_M$, derived from (1) calibration (CLB), (2) MLR and (3) GBM.**



### 3.3 Validation of gridded runoff and ET simulations based on parameter regionalization

To assess the effectiveness of parameter regionalization, we compared the model performance of runoff
and ET simulations with the regionalized parameter sets (MLR and GBM) to that with the calibrated
parameters. Figure 8 and Figure 9 present cumulative density function (CDF) plots of *KGE* values for
runoff and ET simulations in both the calibration (solid lines) and validation (dashed lines) periods over
four climatic zones. The *KGE* values were computed based on the DTVGM-PML simulations using three
parameter sets: (1) grid-by-grid calibration (black lines), (2) MLR generation (blue lines), and (3) GBM
generation (red lines).

Regarding the runoff simulation (Figure 8), median *KGE* values produced by calibrated parameters were
0.783 in humid regions, 0.755 in semi-humid regions, 0.704 in semi-arid regions, and 0.442 in arid regions
for the validation period.  The median *KGE* score based on the MLR method was worse by 0.139 averaged
in four climatic zones than that from simulation using calibrated parameters for the validation period.
While as shown in Figure 8, all the *KGE* values in four regions from GBM parameters were superior to
the MLR parameters. The corresponding difference of *KGE* was 0.023 relative to the calibration for the
validation period.

In contrast to runoff simulation, however, the distributions of *KGE* for ET simulation from the
regionalized parameters were significantly close to that based on the calibrated parameters in each region
as shown in Figure S6. The median *KGE* values from three parameter sets were around 0.68 in humid
regions, 0.74 in semi-humid regions, 0.72 in semi-arid regions, and 0.53 in arid regions for the validation
period. Overall, the performance of ET simulation from regionalization was comparable to that from
calibration satisfactorily.



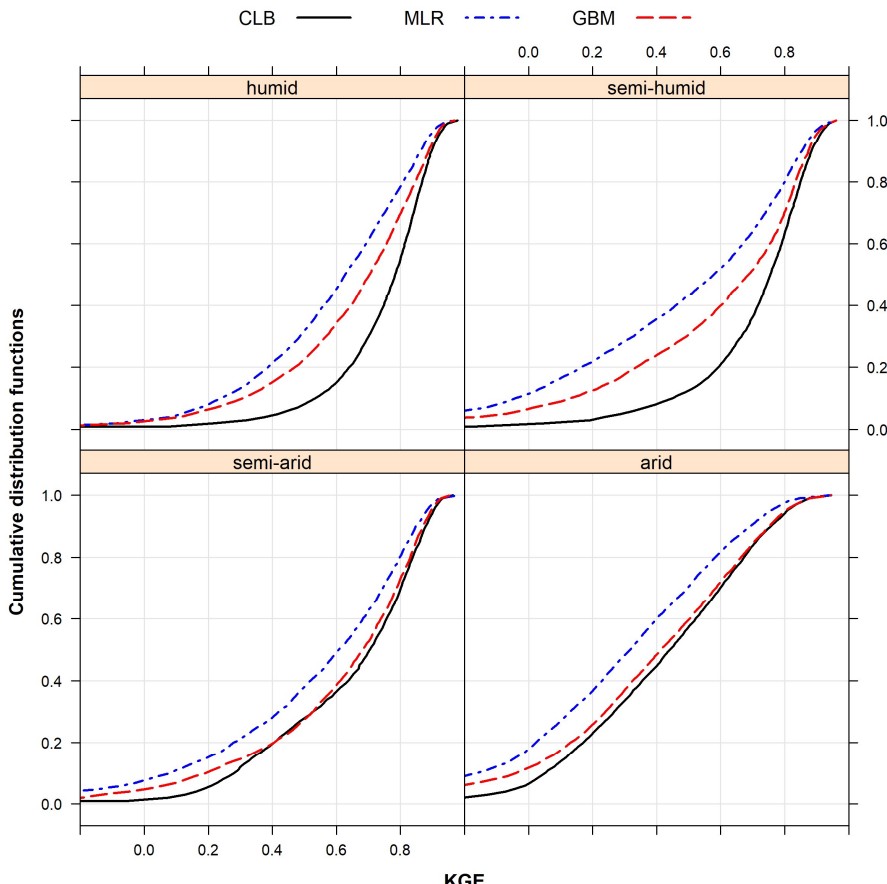

**Figure 8. Cumulative distribution functions (CDFs) of *KGE* for runoff simulation based on three parameter sets (black lines for CLB, blue lines for MLR, red lines for GBM) in the validation period over four climatic zones.**

### 3.4 Validation of watershed streamflow simulations based on parameter regionalization

To give insight into the performance of streamflow simulations based on parameter regionalization, we calculated *NSE*, *KGE*, and *PBIAS* at 31 representative watersheds with both daily and monthly streamflow




validation (Figure 9). According to the Kruskal-Wallis test, there were insignificant differences (p-values < 0.05) in five criteria based on the three parameter sets. Both MLR and GBM performed as well as the calibration with high median scores of *NSE* and *KGE* at both daily (*NSE* was nearly 0.64 and *KGE* was

nearly 0.67) and monthly (*NSE* was nearly 0.84 and *KGE* was nearly 0.78) scales and median *PBIAS* close to zero (around −4.1%). Since the stations were almost located in humid and semi-humid regions (28 of 31 in Figure 3), we could expect that the parameters derived from the regionalization can reasonably generate monthly streamflow with good agreements to the observations (in line with results from Figure 8). We also obtained encouraging results of daily streamflow simulation with satisfying accuracy. In terms

of the three stations in arid and semi-arid regions, the performance of MLR was slightly poorer than that of calibration and GBM, as the daily and monthly *NSE* values, for instance, at Tangnaihai station, were 0.494 and 0.586 compared with the corresponding values of 0.675 and 0.751 for calibration, 0.631 and 0.719 for GBM, respectively.

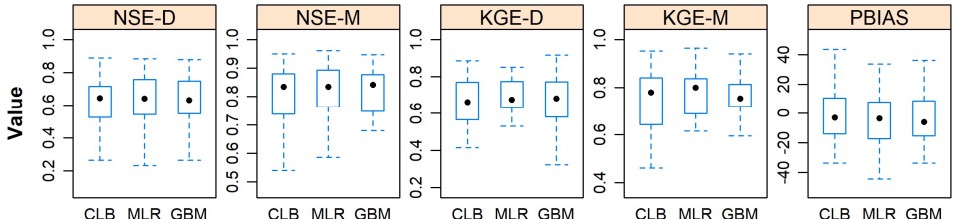

**Figure 9. The model performance statistics of streamflow simulations on 31 hydrological stations based on three parameter sets. 'D' and 'M' denote daily and monthly evaluation, respectively.**

### 3.5 Identification of important factors for model parameters

We further estimated the relative importance of each explanatory variable based on the GBM model, which was determined by averaging the improvement (decrease) in the squared error at each split over all

trees made by each variable, with a range from 0 (least important) and 100 (most important) (Natekin and Knoll, 2013; Xia et al., 2020). Figure 10 presents the relative importance of explanatory variables from the GBM model for all six parameters in four climatic zones and the margin plots containing the mean relative importance for each parameter or each climatic zone.



Generally, we found the slope (slp), saturated moisture content (ths), and elevation (elev) were the most

critical explanatory variables to inform the model parameters (Figure 10g5). Whereas there were several differences among parameters or regions. In humid and semi-humid regions, the terrain properties, including slope and elevation, were likely to determine most model parameters (Figure 10g1 and g2). As for semi-arid and arid regions, most parameters primarily depended on the saturated moisture content that becomes a constraining factor for runoff generation in dry areas. For the three parameters ($g_1$, $g_2$, and $k_s$)

that control surface and subsurface runoff generation, the dominant factors were slope and saturated moisture content (Figure 10a5, b5, and c5). However, the parameters for groundwater recharge ($k_r$) and recession ($k_g$) were mainly controlled by the saturated moisture content, followed by elevation (Figure 10d5 and e5). In terms of the parameter $W_M$, which represents the soil moisture storage capacity, the slope, elevation followed by the saturated hydraulic conductivity were more essential predictors than the

saturated moisture content (Figure 10f5).



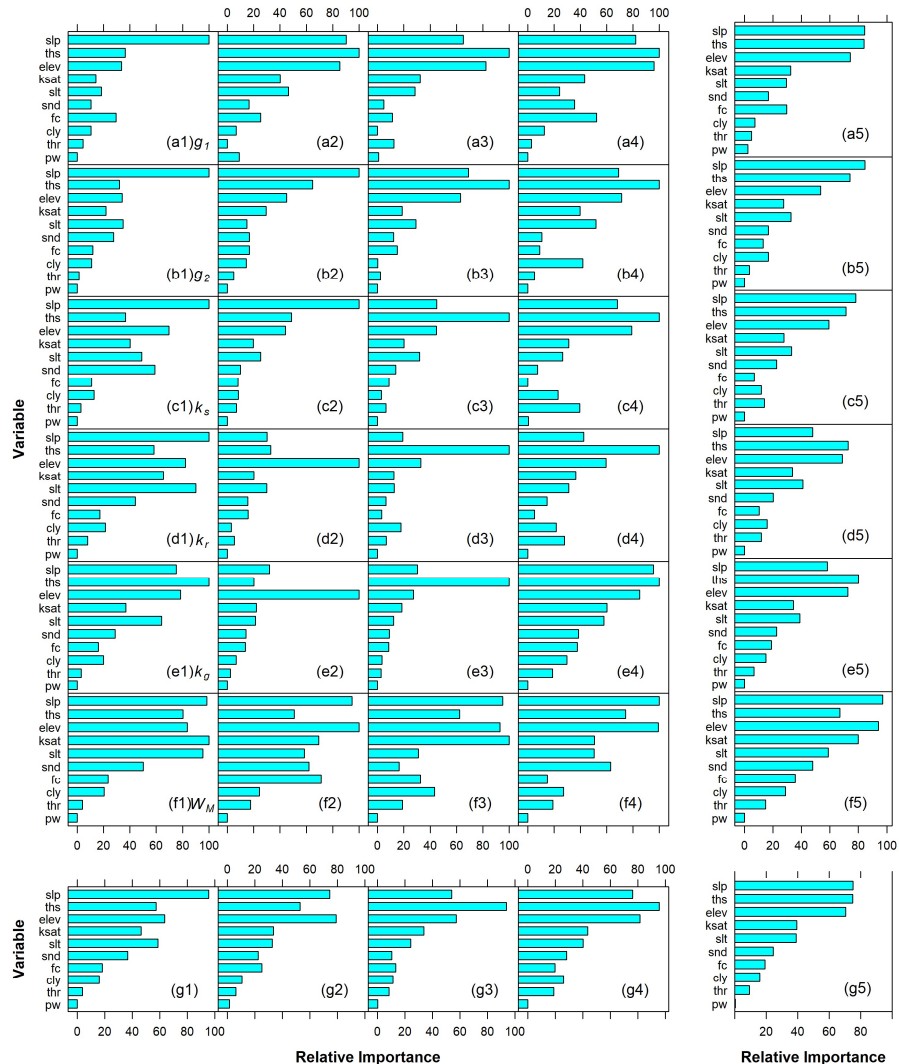

**Figure 10.** Relative importance of explanatory variables from the GBM model for (a) $g_1$, (b) $g_2$, (c) $k_s$, (d) $k_r$, (e) $k_g$, (f) $W_M$ in four climatic zones (1, humid; 2, semi-humid; 3, semi-arid; 4, arid). The mean relative importance over four regions for each parameter is shown in the right column (a5-g5). The mean relative importance over six parameters for each climatic zone is shown in the bottom row (g1-g5). Explanatory variables include elev (Elevation), slp (Slope), snd (Sand content), slt (Silt content), cly (Clay content), fc (Field capacity), pw (Wilting point), thr (Residual moisture content), ths (Saturated moisture content), ksat (Saturated hydraulic conductivity).





## 4 Discussion

### 4.1 Performance of calibration and regionalization for DTVGM-PML

Previous studies have established the effectiveness of the multivariate calibration framework for hydrological models (Bai et al., 2018; Dembélé et al., 2020; Demirel et al., 2018; Finger et al., 2015; Nijzink et al., 2018; Xie et al., 2020). The current study performed a multiple variable calibration strategy to calibrate model parameters in each grid against the reference datasets (VIC runoff and GLEAM ET) during the 15-year calibration period, followed by an independent model validation against the reference

gridded runoff and ET during the 16-year validation period. We also validated the model using observed streamflow at 31 representative hydrological stations in diverse climatic zones. The gridded runoff in each watershed was routed to the corresponding hydrological station using consistent routing parameters. Despite that the streamflow data are commonly used in the calibration of hydrological models (Dembélé et al., (2020), previous studies have explored the potential in model calibration solely against remotely

sensed ET data (without the need for gauged streamflow data) and achieved encouraging results in streamflow simulation (Huang et al., 2020; Zhang et al., 2020). The satisfactory performance in model validation of streamflow suggests the high reliability of the multivariate calibration strategy used in this study.

Hydrological simulation in arid and semi-arid regions is still challenging (Huang et al., 2016; Wheater et

al., 2007; Yang et al., 2019). There appears better performance in runoff simulations in (semi-) humid regions than (semi-) arid regions (Figure 8). Regionalization methods tend to perform more poorly in drier regions, which is expected in common modeling practices (Guo et al., 2021; Parajka et al., 2013). The results support the general knowledge of runoff prediction in different climatic zones, i.e., it is more challenging to achieve good performance in (semi-) arid regions than humid regions. The relatively poor

performance in (semi-) arid regions may be attributed to model structural deficiencies, forcing errors, as well as uncertainties in the reference data for calibration. For example, the large-scale hydrological models may ignore complex processes, like surface-groundwater interactions and channel losses (Oubeidillah et al., 2014). The quality of forcing data (e.g., precipitation) also influences the performance of hydrological models in runoff simulation (Mizukami et al., 2017). Wang et al., (2016) found systematic

overestimates of CMFD precipitation over the Qinghai-Tibetan Plateau. Furthermore, unlike the observed



streamflow, the reference data, such as runoff and ET data, are not the standard actual observed data. This might also explain the better performance in our model validation against observed streamflow at watershed scale with regionalized parameters than calibrated parameters. Zhang et al., (2014) suggested that the data should be used with caution in western China with significant potential uncertainties in hydrologic simulation due to the lack of meteorological observations. And issues in arid regions have always been a challenge for the VIC model (Oubeidillah et al., 2014; Yang et al., 2019). Yang et al., (2017) indicated that the GLEAM ET data showed significantly systematic bias and overestimated the eddy covariance ET measurements at forest sites. We argue that these reanalysis datasets are precious for large-scale calibration of hydrological models in terms of both spatial and temporal dynamics, though it will inevitably introduce uncertainties to a certain degree.

### 4.2 Prediction of model parameters by machine learning methods

Given the complex and nonlinear relationships with high nonstationarity between physical catchment descriptors and model parameters, it is a daunting challenge to develop a robust method for the regionalization of hydrological models (Guo et al., 2021). Machine learning approaches provide a promising tool, as an alternative to conventional linear regression models, to regionalize hydrological model parameters. Machine learning techniques are focused on specific tasks, like classification, regression, and have been widely used in many hydrological issues (Adnan et al., 2019; Huntingford et al., 2019; Lima et al., 2015; Rajaee et al., 2019; Shen, 2018; Yaseen et al., 2015; Zhang et al., 2018). In this study, we built a predictive GBM model for parameter regionalization of DTVGM-PML compared with an MLR model using the terrain and soil properties as explanatory variables. Overall, the GBM outperformed the MLR based on evaluation in three aspects: (i) higher accuracy in reproducing the calibrated parameters as indicated by significantly lower RMSE with GBM against MLR (Figure 6) and more consistent spatial pattern with calibrated parameter values (Figure 7); (ii) better performance in runoff simulations based on parameters generated from GBM than MLR (Figure 8); (iii) comparable results of streamflow validation in 31 watersheds based on regionalization and better performance in several stations in (semi-) arid region by GBM than MLR (Figure 9). Note that the parameters in ET estimation (PML method in DTVGM-PML) were not involved in model calibration. Consequently, the





performance of ET simulation from regionalization was comparable to that from calibration (Figure S6). Taken together, the GBM method, as an ensemble technique, can achieve higher accuracy in parameter
regionalization than MLR (Natekin and Knoll, 2013).

We noticed that the GBM outperformed the MLR in grid-scale runoff simulations but showed insignificant difference in watershed streamflow validation. It is likely due to that (i) the observed watershed streamflow data are independent of the girded runoff data in this study, which could lead to differential modelling performance in these two datasets; (ii) most of the watersheds (i.e., 28 out of 31)
for streamflow validation are within humid regions, where the difference in performance of grid-scale runoff simulations was relatively small, compared with non-humid regions, between regionalized and calibrated parameters; and (iii) the flow routing of grid-scale runoff within a watershed may smooth the heterogeneity in runoff from multiple spatially distributed grid. We suggest that streamflow from diverse watersheds, especially in arid regions, are needed for model validation with parameters derived from
different regionalization approaches. The possible inconsistent results between the evaluation of parameter regionalization and the validation of streamflow also imply the necessity of watershed-scale streamflow validation following parameter regionalization.

The proposed GBM method explored the relationship between model parameters and the terrain and soil attributes, offering a helpful approach to estimate model parameters for hydrological simulations. It can
also achieve satisfactory accuracy, especially in (semi-) humid regions. This study also motivates further investigations including, but not limited to (i) improvement in model structure to better represent hydrological mechanisms in complicated underlying surface conditions and changing environment, (ii) selection of more physical attributes such as vegetation or climatic factors for regionalization.

Although the GBM cannot provide an explicit formula that intuitively links the response variable with
explanatory variables like the MLR, it can also estimate response variables based on explanatory variables. More importantly, the machine learning-based regionalization methods can identify essential driving factors, which develop a primary appraisal of how important terrain and soil properties are for parameters of hydrological models.





### 4.3 Important attributes dominating model parameters

The runoff process is primarily controlled by regional climatic regime, vegetation, land use, topography, and soils (Dunne and Leopold, 1978; Freeze, 1974; Mizukami et al., 2017; Tarboton, 2003). We used the GBM method to predict parameter values by available topographic and soil properties, such as slope, elevation, saturated moisture content, saturated hydraulic conductivity, field capacity, soil texture. Our findings of variable importance using the GBM model quantitatively indicate that the runoff generation

parameters of DTVGM-PML are majorly controlled by slope, saturated moisture content, and elevation. Moreover, the results in different climatic zones show that terrain attributes significantly influence the runoff process in relatively humid regions. At the same time, the saturated moisture content becomes a limiting factor in drier areas.

Prior studies have noted the incredible impact of slope on runoff generation (Akbarimehr and Naghdi,

2012; Chaplot and Le Bissonnais, 2003; Garg et al., 2013; Tarboton, 2003). Steeper slopes lead to faster drainage in aquifers (Beck et al., 2020; POST and JAKEMAN, 1996; Zecharias and Brutsaert, 1988). The slope seems to be the most critical factor for parameters $g_1$, $g_2$, and $k_s$ that control surface and subsurface runoff generation. Garg et al., (2013) investigated that the slope affects surface runoff estimation significantly for Solani watershed using the modified NRCS-CN (natural resources conservation service

curve number) method. Several reports have shown that storm runoff by subsurface flow requires steep, convex slopes and high saturated hydraulic conductivities (Freeze, 1972, 1974; Montgomery and Dietrich, 2002). The parameter $g_1$ presents the surface runoff coefficient when the soil water storage reaches its maximum value, i.e., $W/W_M$=1, and the parameter $g_2$ is the power of the relative soil water storage ($W/W_M$). Steeper slope results in an increase in the value of $g_1$, a decrease in $g_2$ (Figure S5 a1 and b1), and thus

leading to an increase in the amount of surface runoff, which supports the previous findings that runoff amount increases with increasing slope (Chaplot and Le Bissonnais, 2003; Huang, 1995).

Regarding the groundwater recharge and baseflow, the corresponding parameters ($k_r$ and $k_s$) strongly depend on the saturated moisture content. The results are consistent with Chiew and Siriwardena, (2005) who found that the groundwater recharge parameter and baseflow recession parameter of SIMHYD (a

simplified version of the HYDROLOG) model are highly correlated with plant available water holding capacity, a proxy of the soil water storage capacity (McKenzie et al., 2000). While concerning the soil


moisture storage capacity, $W_M$, in DTVGM-PML, the saturated moisture content is likely to be less important than the slope, elevation, and saturated hydrologic conductivity. Note that $W_M$ is different from the saturated moisture content in soil. The latter is equivalent to effective porosity and is simplified as the one-layer value in this study. Whereas the soil moisture state variables in many conceptual hydrological models do not act in the same way as in the real world (Zhuo and Han, 2016). The surface slope is correlated with soil depth (Tesfa et al., 2009), which strongly influences $W_M$. If the soil depth data, more likely to be associated with $W_M$, are available, they should be incorporated in parameter regionalization to obtain more reliable and intelligible results.

**5 Conclusion**

We conducted parameter regionalization for the China-wide DTVGM-PML model using a machine learning technique, the gradient boost machine, compared with the traditional multiple linear regression method. We show that the GBM model is superior to the MLR in predicting model parameters as a function of topographical and soil characteristics due to its significantly lower biases and higher spatial agreement for almost all parameters in four distinct climatic zones. The regionalized parameters also exhibited better spatial coherence relative to the grid-by-grid calibrated parameters. Regarding the model validation of streamflow simulations in 31 hydrological stations, MLR- and GBM-generated parameters could simulate streamflow as accurately as the results with grid-by-grid calibrated parameters (median daily and monthly *NSE* are 0.65 and 0.84, respectively), with the GBM being preferable to MLR in arid regions. This study suggests the watershed-scale streamflow validation following parameter regionalization is necessary due to potential inconsistent results between parameter regionalization evaluation and the streamflow validation. Based on the GBM regionalization results, we found that the slope, saturated moisture content, and elevation are the most important explanatory variables to inform model parameters. Our results indicate that machine learning techniques can be a useful alternative to the conventional regression approach to better predict hydrological model parameters. This is particularly significant for hydrological predictions in ungauged basins. The methods developed and insights gained from this study can also improve the interpretation and prediction of parameters in other large-scale hydrological and environmental models.



**Code and data availability**

The    model    code    used    in    this    study    is    available    via
https://github.com/Songzh101/GBM_HydroModel_Regionalization. The climatic forcing data, land
surface data, elevation data, and evapotranspiration data are available online, as described in Table 1. The
streamflow in the 31 watersheds were obtained from China's Hydrological Year Book. These data are not
publicly available because of governmental restrictions, but can be accessed by contacting the

corresponding author. The soil properties are provided by Liu et al., (2020). The VIC simulations for
China are obtained from Zhang et al., (2014).

**Author contribution**

ZS and GW designed the study. DS, GW and ZS developed the model code and performed the simulations.
ZS prepared the manuscript with contributions from all co-authors. JX, GW, DS, CH, SH contributed to

results discussion and critical review of the manuscript.

**Competing interests**

The authors declare that they have no conflict of interest.

**Acknowledgments**

This work was funded by the National Natural Science Foundation of China (No. 41890823), the National

Key R&D Program of China (Grant Number: 2017YFA0603702). The climatic forcing data set are
provided by National Tibetan Plateau Data Center (http://data.tpdc.ac.cn). The land surface data are
provided by National Earth System Science Data Center, National Science & Technology Infrastructure
of China. (http://www.geodata.cn). We also appreciate the data support from Zhang et al., (2014) and Liu
et al., (2020).




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
