# Peer review of "Regionalization of hydrological model parameters using gradient boosting machine"

_Hydrology and Earth System Sciences, 2021_

## Author Comment (AC2)

**Regionalization of hydrological model parameters using gradient boosting machine**

Zhihong Song, Jun Xia, Gangsheng Wang, Dunxian She, Chen Hu, Si Hong

This discussion paper examines the use of gradient boosting machine learning to model the dependency of model parameters and estimate model parameters for four climate regions in China. The study also enhances the DTVGM model by incorporating the Penman-Monteith-Leuning (PML) equation. The discussion paper is well-presented, the study design is nicely conceived, and the results and discussion are presented fairly, with references to supporting figures, tables or cited literature, where appropriate.

I only have minor comments to enhance what is already a high quality manuscript.

**Response**: We really appreciate your valuable suggestions and comments which can help us improve the quality of our manuscript. We have addressed the comments point-by-point.

(1) In several places, the model is referred to as the "China-wide hydrological model" (see L105). This is somewhat confusing. Is this a formal name for the hydrological model? If so, I would expect to see a citation after the phrase to reference the use of this name. If this is not the formal name, it may be more helpful to say "We ran a hydrological model developed by Beck et al (2020), for country of China in a spatially distributed…" In Section 2.1, the title could be changed to "Application of the hydrological model across China."

Response: Thanks for pointing this out! "China-wide hydrological model" is not a formal name for the model. We will rewrite the sentence to address this issue: "We ran

a hydrological model, i.e., Distributed Time-Variant Gain Model with the Penman-Monteith-Leuning equation (DTVGM-PML) developed in this study, for country of China in a spatially distributed manner"

We will also change the title of Section 2.1 to "Application of the hydrological model across China"

(2) In my reading, there was some confusion about how you were able to compute evaluation metrics for all 15,640 grid cells (see Figure 5 for example). If you know the "truth" for runoff and ET at every grid cell, then why do you need a regionalization model?

**Response**: Hydrologic models often rely on regionalization approaches to transfer information from small to large spatial scale (e.g., from gridcell to subbasin, watershed, and regional scale) (Beck et al., 2020; Mizukami et al., 2017), and from gauged to ungauged catchments (He et al., 2011; Hrachowitz et al., 2013; Pagliero et al., 2019; Parajka et al., 2013).

In this study, though parameters were calibrated and available at each gridcell, the parameter values at around 450 gridcells were not reliable owning to poor model performance (i.e., KEG < 0) (Knoben et al., 2018; Koskinen et al., 2017; Sutanudjaja et al., 2018). Therefore, we only used the calibrated parameters with KGE ≥ 0 (i.e., representing better model performance) for regionalization of parameters. The model performance for 53% of these gridcells (with KGE < 0 prior to regionalization) were improved when we re-ran the model with regionalized parameters. Particularly, the KGE values in 37% of the gridcells (with KGE < 0 prior to regionalization) became positive, indicating a substantive improvement of the modeling performance.

Even though the parameters were well calibrated and available at each gridcell, one might think whether and which topographic and edaphic properties mediate these hydrological parameters. Our machine learning (i.e., gradient boosting machine) based regionalization of parameters enables to estimate six key hydrological parameters using

site-specific characteristics. Following the regionalization of parameters, our results of variable importance quantitatively indicate that the runoff generation parameters are majorly controlled by slope, saturated soil moisture content, and elevation. Moreover, the terrain attributes significantly regulate the runoff processes in relatively humid regions, while the saturated soil moisture content becomes a limiting factor in arid areas. The regionalization of parameters will improve our mechanistic understanding of the runoff generation processes and associated key hydrological parameters under different topographic and edaphic conditions.

(3) Also in Figure 5, it would be helpful to show the comparison of the model performance with and without the PML addition so that one can see in quantifiable terms how the addition of the equation improves the calibration and validation performance.

**Response**: Thanks for the valuable suggestion! We will show the comparison of model performance with and without PML addition in Supplementary Fig. S2 (see Fig. R1). We will also add the following text to describe the results:

"A great deal of previous studies have highlighted the importance of incorporating the vegetation change information into hydrological models to achieve better performance in hydrological simulations (Donohue et al., 2007, 2010; Gerten, 2013; Ivanov et al., 2008; Lei et al., 2014; Thompson et al., 2011). And it has been demonstrated that the PML equation can improve the hydrological simulations under vegetation greening conditions by coupled into hydrological models (Bai et al., 2018; Li et al., 2009; Zhang et al., 2009; Zhou et al., 2013). We also performed the comparison of model performance in hydrological simulation between the original DTVGM (without PML) and DTVGM-PML. As shown in Figure R1, the KGE and PBIAS values of runoff simulation (Fig. R1a, b) based on DTVGM were lower than those from DTVGM-PML. The median KGE and PBIAS values of ET simulation (Fig. R1c, d) were comparable between the two models. However, the lower minimum values in the

boxplots (Fig. R1c, d) indicated DTVGM might produce worse performance (i.e., KGE < 0) than DTVGM-PML. Consequently, DTVGM-PML can help to improve hydrological simulation relative to DTVGM. Additionally, the consideration of vegetation dynamics by the PML equation would improve the mechanistic understanding of hydrological response under vegetation greening, which is lacking in DTVGM."

[Figure]

Figure R1. Comparison of model performance in runoff (a: KGE, and b: PBIAS) and ET (c: KGE, and d: PBIAS) simulation between DTVGM and DTVGM-PML in the calibration and validation periods. KGE denotes the Kling-Gupta efficiency. PBIAS denotes the percent bias".

(4) Section 2.4: You provide an excellent description of the evaluation criteria; however, in your use of the Taylor skill score, could you clarify how the skill is determined for model parameters when you cannot know the true value of the parameters? L207-209 were somewhat confusing. This could be my lack of familiarity with the TSS, but it may be helpful to look over those lines to see if you could improve the explanation there.

**Response**: Thanks for your comments. We will provide more detailed information of the Taylor skill score (TSS) in the revised paper as follows:

"The Taylor skill score, as a comprehensive metric of correlation coefficient, standard deviation, and root mean square error, has been widely used in model evaluation (Mohan and Bhaskaran, 2019; Taylor, 2001).

$$TSS = \frac{4(1+r)^4}{\left(SDR+\frac{1}{SDR}\right)^2 (1+r_0)^4}, \tag{R1}$$

$$r = \frac{\frac{1}{n}\sum_1^n (X-\bar{X})(Y-\bar{Y})}{\sigma_X \sigma_Y}, \tag{R2}$$

$$SDR = \frac{\sigma_X}{\sigma_Y}, \tag{R3}$$

$$\sigma_X = \sqrt{\frac{\sum_1^n (X-\bar{X})^2}{n}}, \ \ \sigma_Y = \sqrt{\frac{\sum_1^n (Y-\bar{Y})^2}{n}}, \tag{R4}$$

where $X$, $Y$ are the calibrated parameter and regionalized parameter in DTVGM-PML (e.g., the runoff generation parameters, $g_1$ and $g_2$), respectively; $\bar{X}$, $\bar{Y}$ are the mean values of $X$ and $Y$, respectively; $\sigma_X, \sigma_Y$ are the spatial standard deviation of calibrated parameter and regionalized parameter, respectively; $r$ represents the spatial correlation coefficient between $X$ and $Y$; $r_0$ is the maximum correlation attainable and usually set to 0.999; $SDR$ is the ratio of $\sigma_X$ to $\sigma_Y$; and $n$ is the total number of values for $X$ (and $Y$).

In this study, we used the grid-scale calibrated parameters as the reference parameters to evaluate the regionalization model performance for estimating each parameter at each grid with TSS. We compared two regionalization methods, the gradient boosting machine (GBM) and the multiple linear regression (MLR). As shown

in Figure R2, GBM obviously outperformed MLR with a higher TSS, suggesting that the GBM-regionalized parameters presented a higher spatial agreement with reference parameter values than the MLR-generated parameters."

[Figure]

Figure R2. Taylor skill scores (TSS) of each parameter generated from the multiple linear regression (MLR) and the gradient boosting machine (GBM). The Taylor skill scores were computed using parameters from all grid cells across China.

(5) In the figure captions, some of the acronyms are spelled out, while others are not. It may be best to spell out all abbreviated words and their abbreviations in the captions so the reader does not have to refer back to the text.

**Response**: Thanks for the suggestion. We will modify corresponding Figure captions.

The figure caption of Fig. 4 "Spatial patterns of mean annual runoff (a) and ET (b) simulations by the DTVGM-PML during 1982–2012" will be modified as "Spatial patterns of mean annual runoff (a) and evapotranspiration (ET) (b) simulations by the

DTVGM-PML during 1982–2012.".

[revised manuscript text omitted]

---

## Author Response (AR1)

Authors' responses to comments

**Regionalization of hydrological model parameters using gradient boosting machine**

Zhihong Song, Jun Xia, Gangsheng Wang, Dunxian She, Chen Hu, Si Hong

We really appreciate the editor's and anonymous reviewers' valuable suggestions and comments on the manuscript. All suggestions are helpful to improve this manuscript. We have addressed the comments point-by-point and revised the manuscript accordingly.

**Responses to Referee #1's comments**

The manuscript is interesting and clearly written. I have a single comment; in particular, what are the practical reasons for regionalizing the parameters of the hydrological model using regression algorithms, given that available data cover uniformly the region of the study; therefore, regionalization is not needed in my opinion (parameters are already available).

**Response**: We sincerely appreciate the referee's positive comments on the manuscript. Our response to the reviewer's concern on regionalizing the model parameters is as follows and has been revised in the discussion part of the manuscript (Sec. 4.4, Page 29).

"Hydrologic models often rely on regionalization approaches to transfer information from small to large spatial scale (e.g., from gridcell to subbasin, watershed, and regional scale) (Beck et al., 2020; Mizukami et al., 2017), and from gauged to ungauged catchments (He et al., 2011; Hrachowitz et al., 2013; Pagliero et al., 2019; Parajka et al., 2013).

In this study, though parameters were calibrated and thus available at each gridcell,

the parameter values at around 450 gridcells were not reliable owning to poor model performance (i.e., KEG < 0) (Knoben et al., 2018; Koskinen et al., 2017; Sutanudjaja et al., 2018). Therefore, we only used the calibrated parameters with KGE ≥ 0 (i.e., representing better model performance) for regionalization of parameters. The model performance for 53% of these gridcells (with KGE < 0 prior to regionalization) were improved when we re-ran the model with regionalized parameters. Particularly, the KGE values in 37% of the gridcells (with KGE < 0 prior to regionalization) became positive, indicating a substantive improvement of the modeling performance.

Even though the parameters were well calibrated and available at each gridcell, one might think whether and which topographic and edaphic properties mediate these hydrological parameters. Our machine learning (i.e., GBM) based regionalization of parameters enables to estimate six key hydrological parameters using site-specific characteristics. Following the regionalization of parameters, our results of variable importance quantitatively indicate that the runoff generation parameters are majorly controlled by slope, saturated soil moisture content, and elevation. Moreover, the terrain attributes significantly regulate the runoff processes in relatively humid regions, while the saturated soil moisture content becomes a limiting factor in arid areas. The regionalization of parameters will improve our mechanistic understanding of the runoff generation processes and associated key hydrological parameters under different topographic and edaphic conditions."

**Responses to Referee #2's comments**

This discussion paper examines the use of gradient boosting machine learning to model the dependency of model parameters and estimate model parameters for four climate regions in China. The study also enhances the DTVGM model by incorporating the Penman-Monteith-Leuning (PML) equation. The discussion paper is well-presented, the study design is nicely conceived, and the results and discussion are presented fairly, with references to supporting figures, tables or cited literature, where appropriate.

I only have minor comments to enhance what is already a high quality manuscript.

**Response**: We sincerely appreciate the referee's positive comments on the manuscript. All the comments and suggestions have been replied to below and have been addressed in the revision.

(1) In several places, the model is referred to as the "China-wide hydrological model" (see L105). This is somewhat confusing. Is this a formal name for the hydrological model? If so, I would expect to see a citation after the phrase to reference the use of this name. If this is not the formal name, it may be more helpful to say "We ran a hydrological model developed by Beck et al (2020), for country of China in a spatially distributed…" In Section 2.1, the title could be changed to "Application of the hydrological model across China."

**Response**: Thanks for pointing this out! "China-wide hydrological model" is not a formal name for the model. We have rewritten the sentence to address this issue: "We ran a hydrological model, i.e., Distributed Time-Variant Gain Model with the Penman-Monteith-Leuning equation (DTVGM-PML) developed in this study, for country of China in a spatially distributed manner" [Line 106, Page 5]. We have also changed the title of Section 2.1 to "Application of the hydrological model across China". Other corresponding information has been revised [Line 103, Page 5; Line 130, Page 6; Line 549, Page 30].

(2) In my reading, there was some confusion about how you were able to compute evaluation metrics for all 15,640 grid cells (see Figure 5 for example). If you know the "truth" for runoff and ET at every grid cell, then why do you need a regionalization model?

**Response**: Our response to the reviewer's concern on regionalizing the model parameters is as follows and has been revised in the discussion part of the manuscript [Sec. 4.4, Page 29].

"Hydrologic models often rely on regionalization approaches to transfer information from small to large spatial scale (e.g., from gridcell to subbasin, watershed, and regional scale) (Beck et al., 2020; Mizukami et al., 2017), and from gauged to ungauged catchments (He et al., 2011; Hrachowitz et al., 2013; Pagliero et al., 2019; Parajka et al., 2013).

In this study, though parameters were calibrated and thus available at each gridcell, the parameter values at around 450 gridcells were not reliable owning to poor model performance (i.e., KEG < 0) (Knoben et al., 2018; Koskinen et al., 2017; Sutanudjaja et al., 2018). Therefore, we only used the calibrated parameters with KGE $\geq$ 0 (i.e., representing better model performance) for regionalization of parameters. The model performance for 53% of these gridcells (with KGE < 0 prior to regionalization) were improved when we re-ran the model with regionalized parameters. Particularly, the KGE values in 37% of the gridcells (with KGE < 0 prior to regionalization) became positive, indicating a substantive improvement of the modeling performance.

Even though the parameters were well calibrated and available at each gridcell, one might think whether and which topographic and edaphic properties mediate these hydrological parameters. Our machine learning (i.e., GBM) based regionalization of parameters enables to estimate six key hydrological parameters using site-specific characteristics. Following the regionalization of parameters, our results of variable importance quantitatively indicate that the runoff generation parameters are majorly

controlled by slope, saturated soil moisture content, and elevation. Moreover, the terrain attributes significantly regulate the runoff processes in relatively humid regions, while the saturated soil moisture content becomes a limiting factor in arid areas. The regionalization of parameters will improve our mechanistic understanding of the runoff generation processes and associated key hydrological parameters under different topographic and edaphic conditions."

(3) Also in Figure 5, it would be helpful to show the comparison of the model performance with and without the PML addition so that one can see in quantifiable terms how the addition of the equation improves the calibration and validation performance.

**Response**: Thanks for the valuable suggestion! We have shown the comparison of model performance with and without PML addition in Supplementary Fig. S2 (see Fig. R1). We have also added the following text to describe the results:

"A great deal of previous studies have highlighted the importance of incorporating the vegetation change information into hydrological models to achieve better performance in hydrological simulations (Donohue et al., 2007, 2010; Gerten, 2013; Ivanov et al., 2008; Lei et al., 2014; Thompson et al., 2011). Additionally, it has been demonstrated that coupling the PML equation into hydrological models can improve the hydrological simulations under vegetation greening conditions (Bai et al., 2018; Li et al., 2009; Zhang et al., 2009; Zhou et al., 2013)." [Line 135–140, Page 7]

"We also performed the comparison of model performance in hydrological simulation between the original DTVGM (without PML) and DTVGM-PML. As shown in Figure R1, the KGE and PBIAS values of runoff simulation (Fig. R1a, b) by DTVGM were lower than those from DTVGM-PML. The median KGE and PBIAS values of ET simulation (Fig. R1c, d) were comparable between the two models. In summary, DTVGM-PML can help to improve hydrological simulations relative to DTVGM. Additionally, the consideration of vegetation dynamics by the PML equation

in DTVGM-PML would improve the mechanistic understanding of hydrological response under vegetation greening, which is lacking in DTVGM." [Line 278–285, Page 14,15]

[Figure]

Figure R1. Comparison of model performance in runoff (a: KGE, and b: PBIAS) and ET (c: KGE, and d: PBIAS) simulation between DTVGM and DTVGM-PML in the calibration and validation periods. KGE denotes the Kling-Gupta efficiency. PBIAS denotes the percent bias".

(4) Section 2.4: You provide an excellent description of the evaluation criteria; however, in your use of the Taylor skill score, could you clarify how the skill is determined for

model parameters when you cannot know the true value of the parameters? L207-209 were somewhat confusing. This could be my lack of familiarity with the TSS, but it may be helpful to look over those lines to see if you could improve the explanation there.

**Response**: Thanks for your comments. We have provided more detailed information of the Taylor skill score (TSS) in the revised paper as follows:

"The Taylor skill score, as a comprehensive metric of correlation coefficient, standard deviation, and root mean square error, has been widely used in model evaluation (Mohan and Bhaskaran, 2019; Taylor, 2001)." [Line 214–216, Page 10]

$$TSS = \frac{4(1+r)^4}{\left(SDR+\frac{1}{SDR}\right)^2(1+r_0)^4}, \tag{R1}$$

$$r = \frac{\frac{1}{n}\sum_{i=1}^{n}(X_i-\bar{X})(Y_i-\bar{Y})}{\sigma_X\sigma_Y}, \tag{R2}$$

$$SDR = \frac{\sigma_X}{\sigma_Y}, \tag{R3}$$

$$\sigma_X = \sqrt{\frac{\sum_{i=1}^{n}(X_i-\bar{X})^2}{n}}, \ \sigma_Y = \sqrt{\frac{\sum_{i=1}^{n}(Y_i-\bar{Y})^2}{n}}, \tag{R4}$$

where $X$, $Y$ are the calibrated parameter and regionalized parameter in DTVGM-PML (e.g., the runoff generation parameters, $g_1$ and $g_2$), respectively; the subscript $i$ denotes the $i$th sample of the gridded parameter; $\bar{X}$, $\bar{Y}$ are the mean values of $X$ and $Y$, respectively; $\sigma_X$, $\sigma_Y$ are the spatial standard deviation of calibrated parameter and regionalized parameter, respectively; $r$ represents the spatial correlation coefficient between $X$ and $Y$; $r_0$ is the maximum correlation attainable and usually set to 0.999; $SDR$ is the ratio of $\sigma_X$ to $\sigma_Y$; and $n$ is the total number of values for $X$ (and $Y$)." [Line 227–237, Page 11]

"We also calculated the Taylor skill scores (TSS) with the grid-scale calibrated parameters as the reference parameters to evaluate the regionalization model performance for estimating each parameter at each grid. As shown in Figure S6 (Fig. R2), GBM obviously outperformed MLR with a higher TSS, suggesting that the GBM-regionalized parameters presented a higher spatial agreement with reference parameter

values than the MLR-generated parameters." [Line 311–315, Page 16,17]

[Figure]

Figure R2. Taylor skill scores (TSS) of each parameter generated from the multiple linear regression (MLR) and the gradient boosting machine (GBM). The Taylor skill scores were computed using parameters from all grid cells across China.

(5) In the figure captions, some of the acronyms are spelled out, while others are not. It may be best to spell out all abbreviated words and their abbreviations in the captions so the reader does not have to refer back to the text.

**Response**: Thanks for the suggestion. We have modified corresponding Figure captions.

The figure caption of Fig. 4 "Spatial patterns of mean annual runoff (a) and ET (b) simulations by the DTVGM-PML during 1982–2012" has been modified as "Spatial patterns of mean annual runoff (a) and evapotranspiration (ET) (b) simulations by the DTVGM-PML during 1982–2012.".

The figure caption of Fig. 5 "Model performance of runoff (KGE: a, and PBIAS: b) and ET (KGE: c, and PBIAS: d) simulations in calibration and validation periods. The boxplot was generated using data from a total of 15640 grid cells over China" has been modified as "Model performance of runoff (KGE: a, and PBIAS: b) and evapotranspiration (ET) (KGE: c, and PBIAS: d) simulations in calibration and validation periods. The boxplot was generated using data from a total of 15640 grid cells over China. KGE denotes the Kling-Gupta efficiency. PBIAS denotes the percent bias".

The figure caption of Fig. 6 has been modified as "Performance evaluation of the multiple linear regression (MLR) and the 
[revised manuscript text omitted]